# Novel Microwave Synthesis of Copper Oxide Nanoparticles and Appraisal of the Antibacterial Application

**DOI:** 10.3390/mi14020456

**Published:** 2023-02-15

**Authors:** Rajaram Rajamohan, Chaitany Jayprakash Raorane, Seong-Cheol Kim, Sekar Ashokkumar, Yong Rok Lee

**Affiliations:** 1School of Chemical Engineering, Yeungnam University, Gyeongsan 38541, Republic of Korea; 2Plasma Bioscience Research Center, Kwangwoon University, Seoul 01897, Republic of Korea

**Keywords:** metal oxide nanoparticles, microwave synthesis, X-ray photoelectron spectroscopy, bacterial pathogens, seed germination

## Abstract

The exceptional characteristics of bio-synthesized copper oxide nanoparticles (CuO NPs), including high surface-to-volume ratio and high-profit strength, are of tremendous interest. CuO NPs have cytotoxic, catalytic, antibacterial, and antioxidant properties. Fruit peel extract has been recommended as a valuable alternative method due to the advantages of economic prospects, environment-friendliness, improved biocompatibility, and high biological activities, such as antioxidant and antimicrobial activities, as many physical and chemical methods have been applied to synthesize metal oxide NPs. In the presence of apple peel extract and microwave (MW) irradiation, CuO NPs are produced from the precursor CuCl_2_. 2H_2_O. With the help of TEM analysis, and BET surface area, the average sizes of the obtained NPs are found to be 25–40 nm. For use in antimicrobial applications, CuO NPs are appropriate. Disk diffusion tests were used to study the bactericidal impact in relation to the diameter of the inhibition zone, and an intriguing antibacterial activity was confirmed on both the Gram-positive bacterial pathogen *Staphylococcus aureus* and Gram-negative bacterial pathogen *Escherichia coli*. Moreover, CuO NPs did not have any toxic effect on seed germination. Thus, this study provides an environmentally friendly material and provides a variety of advantages for biomedical applications and environmental applications.

## 1. Introduction

Nanotechnology, which deals with the engineering of matter with a minimum of one dimension spanning the size of less than 100 nm, is a rapidly expanding area [1]. In contrast to their bulk analogs, nanoparticles behave better for many applications due to their high surface-to-volume ratio and greater surface reactivity [2,3]. The high electrical conductivity, chemical stability, large bandgap, and high transmittance of the metal oxide nanoparticles [4,5,6] increase their suitability as a potential material for a number of applications, including self-cleaning technology, sensor fabrication, data storage devices, and photocatalysis [7,8,9]. However, because they eventually get into our environment, their use needs to be carefully considered. Therefore, the investigation is essential to determining how these nanoparticles affect plants.

NPs have been used in numerous industrial and consumer goods during the past few decades. Different sectors are creating unique NPs to enhance their services and goods as a result of the growing use of NPs in commercial items. NPs may be discharged into the environment as a result of some of the NP-intensive industries [10]. A small number of the many NPs are employed on a big scale and may end up in the environment [10]. The NPs have the ability to contaminate the environment through a variety of activities, including inappropriate industrial waste management and improper user product disposal. A wide variety of materials are covered by NPs [11], but only a few of them are widely used at the moment, putting the environment in danger of exposure. As they are most frequently employed in industry, metal nanoparticles (silver, copper, aluminum, nickel, and iron) and metal oxide nanoparticles (titanium dioxide, zinc oxide, cerium dioxide, and copper oxide) are mostly investigated for their effects on various plants [12]. CuO NP has been demonstrated to increase plants’ ability to produce reactive oxygen species (ROS) [13,14,15]. In plants treated with NPs, various antioxidant molecules were seen to be greatly enhanced, showing that plants had activated the protective mechanism [16].

Due to their distinct physicochemical and biological characteristics, CuO NPs have received a lot of attention among these metal oxide NPs. Numerous physical and chemical techniques can be used to create metal oxide nanoparticles (MO NPs) [17,18,19,20]. However, the creation of NPs must be done in a way that is environmentally friendly [21,22,23]. The employment of environmentally friendly biological techniques is preferred. As a result, they are regarded as “green” methods. Recently, fruit peels were used to make metal or metal oxide nanoparticles [24,25,26,27]. Due to its sustainability, accessibility, and simplicity, CuO NPs’ photosynthesis has attracted more attention lately [28]. The objective of this study was to characterize CuO NPs in an eco-friendly manner and then show their antibacterial activity.

In this study microwave irradiation process was used to make CuO NPs from the CuCl_2_ precursor and their toxic effect on seed germination of *Raphanus raphanistrum* seeds has been documented. The synthesized NPs are characterized with the help of analytical techniques like XRD, FE-SEM, HR-TEM, XPS, BET, and Raman spectral analysis. To the best of our knowledge, this is the first investigation examining the impact of green synthetic CuO NPs on the seed germination of *R. raphanistrum* seeds and in vitro antibacterial potency of CuO NPs against *Staphylococcus aureus* and *Escherichia coli*. This study provides a practical, affordable, and safe method for the creation of inorganic nanoparticles and their use in effective nano fertilizers and other biomedical applications.

## 2. Materials and Methods

The details of materials and methods of synthetic CuO NPs (A scheme for synthe-sizing NPs is shown in Appendix A), the instruments used for the characterization of NPs, Appendix A provides an overview of the proposed mechanism for making CuO NPs, and the synthesized CuO NPs are tested for cytotoxicity with MDA MB 231 cell line (Appendix A) are provided in the Appendix A.

### 2.1. Antibacterial Activity and Safety Assessment of CuO NPs

#### 2.1.1. In-Vitro Antibacterial Efficacy

The agar well diffusion method is employed to evaluate the antibacterial activity of CuO NPs [29]. The microorganisms *E. coli* (ATCC 43895) and *S. aureus* (ATCC 6538) are employed in this study. In a nutshell, sterile Mueller–Hinton agar (MHA) plates are loaded with 50.0 L of CuO NPs at different concentrations (50.0, 100.0, and 200.0 g/mL (*w/v*)) and punctured with a 7 mm-diameter cork borer. Overnight cultures of each bacterial strain were distributed on the MHA plates at 0.5 McFarland standard. After the incubation period, a Vernier caliper is used to determine the radius of the inhibitory zone. In this investigation, Mueller–Hinton broth media was used to determine minimum inhibitory concentrations (MICs) of synthesized CuO NPs according to the Clinical Laboratory Standards Institute (CLSI) for bacteria. MIC was defined as the lowest concentration that inhibited cell growth. Briefly, freshly grown cells were diluted for the optimum size of inoculum for MICs and treated with various concentrations of CuO NPs in a 96-well microtiter plate. At least two independent cultures are used in experiments to ensure accuracy and reproducibility.

#### 2.1.2. Seed Germination Toxicity Assessment of CuO NPs

Using Murashige and Skoog agar plates, as previously described [30], researchers looked at the effect of CuO NPs on seed germination using *Raphanus raphanistrum* seeds that had been soaked overnight. The seeds were sterilized with 1 mL of 100% ethanol before the experiment, and they were then submerged in a solution of 3% sodium hypochlorite for 15 min. The seeds were subsequently put on agar plates (0.86 g/L Murashige and Skoog medium) that contained both CuO NPs at 0–500 µg/mL and 0.7% bacto-agar. The plates were then imaged after being incubated for seven days at room temperature.

## 3. Results and Discussion

### 3.1. XRD Analysis of Green Synthetic CuO NPs

An effective analytical method to learn more about the crystalline peaks in metal oxide NPs was XRD pattern analysis [31,32,33]. The XRD patterns of green synthetic CuO NPs are displayed in Figure 1. The patterns for both NPs appear to have shown a significant peak at Miller indices (021), (110), (002), (111), (200), (130), (202), (020), (002), and (113), respectively, at 30.2, 31.5, 35.0, 37.5, 39.0, 40.8, 48.0, 49.5, 58.0, and 61.5° [34]. According to the strong peaks in the XRD patterns, the CuO NPs were crystalline and in the monoclinic phase. It was discovered that the lattice parameters a, b, and c were 4.68, 3.41, and 5.08, respectively. According to the well-known Scherrer equation, the average crystal size for the NPs was discovered to be 41.6 nm [35].
*D* = (*Kλ*)/*(β*.cos*θ*)(1)
where *D* is the crystallite size (in nanometers), *K* is the Scherrer’s constant, commonly taken to be 0.9, *β* is the whole width half maximum (in radians), *λ* is the wavelength of the Cu *K* radiation (1.54 Å), and *θ* is the Bragg angle (in degrees).

### 3.2. FE-SEM Analysis of Green Synthetic CuO NPs

CuO NPs’ surface morphology was examined visually and analyzed using FE-SEM images [36,37]. Figure 2A,B show an FE-SEM picture of both NPs at various magnifications. The CuO NPs had a particle-like structure and a consistent shape. CuO NPs exhibit homogenous distributions with particle sizes ranging from 45.0 to 110.0 nm. Images exhibit clusters that have agglomerated together and some square-shaped particles. Clusters of material-like particles can also develop with stabilized NPs when they are positioned closely together. The peel extract decreased and stabilized NPs, allowing for their redispersion [38]. A peel extract restricts flocculation and clustering to regulate particle size distribution. Apple peels are a successful stabilizing agent for the fabrication of NPs in small sizes. As a result, procedures like particle growth, impurity adsorption, and aggregation were responsible for determining the structure of NPs [39]. The chemical composition and particle dispersion of both NPs on the entire surface are confirmed by the EDX data. Figure 2C,D show that Cu and O can be found in the NPs. Cu La, Cu Ka, and Cu Kb representations for the copper reveal that the strong signals are seen around 0.91, 8.04, and 8.92 keV, respectively, and for oxygen (O Ka), about 0.54 keV [40]. Figure 2C,D also include information on the weight percentage and atomic percentage of NPs. The carbon material was not taken into account in the NP composition estimate because it was coated with NPs. These outcomes proved that microwave synthesis can produce a CuO structure in about five minutes.

### 3.3. HR-TEM Analysis of Green Synthetic CuO NPs

The particle size and crystallinity of the NPs are revealed by the TEM pictures and their SAED patterns [41]. HR-TEM pictures of the obtained NPs with 31 nm scale bars are shown in Figure 3A–E,G–K. The stone NPs in these photos have a tight variation of particle sizes and an average diameter of 40.2 4.0 nm. ImageJ software was used to analyze particle size distributions. The linkages, spherical nature, and NP aggregation in Figure 3 closely resemble the FESEM pictures. Figure 3F,L show that the SAED pattern is indexed to planes (021), (110), (002), (111), (200), (130), (202), (020), (002), (021), and (002), respectively, which represent the FCC crystalline structure of CuO. (113). They are stone structures and agglomerated, according to microscopic visualization with FE-SEM and HR-TEM morphological characterization of the NPs.

### 3.4. XPS Analysis of Green Synthetic CuO NPs

XPS analysis, a potent surface-sensitive method for determining CuO oxidation state and chemical composition in NPs, has been used to evaluate the green synthetic CuO NPs [42]. The C 1s peak, which formed at a binding energy of 284.60 eV, is used as a reference for standardizing all binding energies. The peaks of the XPS wide-scan spectra for both NPs are connected to the elements Cu, C, and O, as shown in Figure 4A,E. Figure 4B–D,F–H show high-resolution measurements of the XPS spectra of Cu 2p, C 1s, and O 1s (core XPS spectra). Cu 2ps’ core level spectra or narrow energy range shows a dominating peak at the Cu 2p_3/2_ atom’s stronger binding side and an increase in the main peak’s binding energy, both of which point to an unfilled Cu 3d_9_ shell. The discovery of an unfilled Cu 3d_9_ shell in the CuO sample [43] further supports the discovery of Cu^2+^. Additionally, Cu 2p_3/2_ of CuO is responsible for the peaks at 954.48 and 954.28 eV for the NPs of CC 100 and 110, respectively, in the core level spectra of Cu 2p (deconvolution of CuO NPs, Figure 4B,F). Similar to this, Cu 2p_1/2_ of CuO is responsible for the peaks at 934.48, and 934.48 eV for the NPs of CC 100, and 934.48 eV for the NPs of CC 110 (Table 1), respectively, in the core level spectra of Cu 2p (deconvolution of CuO NPs, Figure 4B,F).

The NPs of CC 100 and CC 110 each have a single component at bonding energies of 531.08 and 531.08, respectively, according to the Gaussian–Lorentzian fit of O1s (Figure 4C,G). One can conclude that the peaks represent the binding energy for oxygen vacancies or flaws in the CuO NPs’ surrounding environment [44]. Figure 4D,H display a high-resolution spectrum of carbon (C 1s), which supports the appearance of the two reference peaks. The first peak is at 284.18, and the second one is at 284.28 eV for the NPs of CC 100 and CC 110, respectively. These higher energy peaks indicate adventitious carbon containing the C-C bond at 288.38 and 288.88 eV for the NPs of CC 100 and 110, respectively. The two peaks of the C 1s spectrum are referred to as contamination of adventitious carbon and serve as a charge reference for the XPS spectra on the surface of nanoparticles (NPs). There was no potential for residual nitrogen in the precursor, as seen by the NPs’ XPS spectra. The structural stability of CuO NPs was confirmed by evaluating the XPS spectrum.

### 3.5. BET Surface Area Analysis

The most effective quantitative model for calculating surface area is the Brunauer– Emmett–Teller (BET) model. Using N_2_ gas adsorption and BET surface area analysis, the pore size distribution and surface area of CuO NPs are investigated. The type IV isotherm obtained for both the CuO NPs (CC 100 and CC 110) and the adsorption-desorption curve are presented in Figure 5A,E. The presence of the mesoporous character of the produced NPs is confirmed by the hysteresis loop within the relative pressure (P/P_0_), which varies from 0.8 to 0.9 [45]. Most often, mesoporous is the term used to describe the pore size that results in a Type IV isotherm. The hysteresis loop is the Type IV isotherm’s defining trait. However, the amount adsorbed is always more at any point on the desorption curve than on the adsorption curve, regardless of the actual shape of the loop, which changes from one adsorption system to the next [46]. H1 loops, which are frequently attained for agglomerates or compacts of spheroidal particles of uniform size and array, are provided by CC 100. H3 is produced by CC 110, which also has adsorbents and pores with a slit form [46,47]. By using the typical multi-point BET, the surface areas of both NPs were determined to be 12.9758 and 2.4368 m^2^/g, respectively (Figure 5B,F). By using the BJH desorption method, the pore size distribution of the generated NPs was examined [48,49]. As can be seen in Figure 5C,D,G,H, the pore size values for the NFs of CC 100 and CC 110 were determined to be 10.64 and 17.08 nm, respectively. Table 2 combines the outcomes of pore size and surface area. Because of this, the pore size indicates that the NPs are mesoporous, which was consistent with the results of other characterization techniques, the XRD, FESEM, and HRTEM images.

### 3.6. Raman Spectral Analysis of Green Synthetic CuO NPs

The main method used to determine the vibrations of metal oxide NPs and local atomic arrangements, and to examine their structural properties was Raman spectroscopy [50,51]. Additionally, it can be used to gauge how crystalline materials like NPs are. CuO NPs’ Raman spectra were displayed in Figure 6. A prominent peak was visible in both spectra at a wavelength of 285.0 cm^−1^, which matched the Ag mode of vibration. The Bg mode of vibration was indicated by the shoulder-like peaks that occurred at 310.0 cm^−1^ [51,52]. The Bg mode of vibration was represented by the medium peak (blue oval in the figure) that appeared for both NPs at 610.0 cm^−1^ [53,54]. Only oxygen atoms have a dislocation shift in the b-axis of A_g_ and B_g_’s Raman modes. By reducing particle size, it was possible to change a Raman shift and bandwidth [52].

### 3.7. FT-IR Spectral Analysis of Green Synthetic CuO NPs

To determine the structural and chemical characteristics of the generated metal oxides, the impacts of the peel extract employed in the synthesis of NPs were examined by FT-IR analysis [53,54]. FT-IR spectra between 400.0 and 4000.0 cm^−1^ were taken (Figure 7). CuO NPs feature a peak that was linked to hydroxyl group stretching, and it is located between 3440 and 3316 cm^−1^. Because of a bending O-H, NPs have a peak that resembles a shoulder at 1645 cm^−1^. The C-O imbalance in NPs was responsible for another little hump at 1386 cm^−1^. The C-O symmetry explained a smaller hump in the spectra at 1124 cm^−1^. The presence of Cu-O bonds was indicated by a strong peak at 538 cm^−1^. The Au mode and Bu modes of CuO occurred at 435 cm^−1^ and 489 cm^−1^, respectively, and were two of the metal’s distinctive bands [55]. The NPs of CC 100 and CC 110 exhibit the high-frequency mode at 590 cm^−1^ and 580 cm^−1^. Cu-O stretching vibrations in the (101) direction have been suggested as the cause [56]. The FT-IR investigation thus validates the monoclinic structure of the pure phase CuO.

### 3.8. DRS Analysis of Green Synthetic CuO NPs

The DRS spectrum of pure CuO NPs is seen in Figure 8. UV-VIS-NIR spectrophotometers were used to measure DRS curves. NPs have absorption bands in the 365.0 nm (blue oval I) range [25,57]. A peak and shoulder were seen at 560.0 nm (blue oval II), indicating the presence of CuO on the surface of NPs. Additionally, the reflectance was weak in both the UV and visible spectrums (200.0–800.0 nm). Additionally, it revealed details regarding the regions’ higher absorption rates given their low transmittance.

### 3.9. TGDTA Analysis of CuO NPs

The thermal behavior of the green synthetic CuO NPs in which a weight loss occurs according to the temperature can be seen with the analysis of TGDTA curves [58]. Figure 9 gives the CuO NPs’ TGDTA curves. In the temperature ranges of 50.0–110.0, 245.0–290.0, 390.0–420.0, and 465.0–660.0 °C, there are four stages of weight loss that can be seen. Due to the removal of water molecules adsorbed on NP surfaces, a very low and progressive weight loss of 1.0% is seen in the first stage. This stage, which results in a modest weight loss, reveals that none of the NFs contain much moisture. The phytoconstituents produced during those temperature fluctuations account for the remaining weight losses in the next stages. The stabilizing chemicals that were used to coat the NPs may be removed at this point, and there is a slight but considerable weight loss at the end of NPs. Both NPs displayed a weight decrease of about 50%. The release of moisture content, and phytoconstituents from the surface of NPs by which they are physically adsorbed [59], as well as the release of peel extract from the surface due to the desorption process held on the surface of NPs, are thought to account for only 60% of the NPs’ total weight loss. The exothermic peaks in the DTA curves of both NPs at 270.0, 415.0, and 600.0 °C demonstrated that the NPs discharged energy into the surrounding environment.

### 3.10. Antibacterial Efficacy of CuO NPs

CuO NPs demonstrate effective activity on both the Gram-positive and Gram-negative bacterial pathogens that are tested. NPs are effective against *E. coli* and *S. aureus*, with MIC values of 25.0 μg/mL and 50 μg/mL, respectively. Our outcomes revealed that the synthesized CuO NPs have a superior antibacterial effect against Gram-negative bacterial strain *E. coli* (ATCC 43895) compared to Gram-positive *S. aureus* (ATCC 6538). This might be because Gram-positive bacteria has a strong cell wall, whereas Gram-negative bacteria has a thin cell wall. Thus, it is possible that CuO NPs easily penetrate to the cell membrane of Gram-negative bacteria and cause damage to the cell [60]. The agar diffusion test, which is also used to evaluate the antibacterial activity of CuO NPs, revealed a distinct zone for action against both bacteria. After the treatment, the sizes of the inhibitory zones were combined in Table 3, and Figure 10 displayed sample photographs. The outcomes showed that, in relation to the concentration of CuO NPs, the zone of inhibition expanded dramatically. The concentrations of NPs and the zone of inhibition were inversely correlated. The zone of inhibition for *E. coli* and *S. aureus* at 200 µg/mL of CC 100 was found to be 23.1 ± 2.3 mm and 16.0 ± 1.0 mm, respectively. Additionally, CC 110 demonstrated a similar pattern of inhibition zones in *E. coli* and *S. aureus* following the treatment; these inhibition zones were measured to be, respectively, 23.2 ± 1.3 mm and 16.1 ± 0.9 mm at 200 µg/mL. It has been demonstrated that CuO NPs are highly effective against a variety of bacterial strains [61]. Additionally, against *E. coli* and *S. aureus*, both CuO NPs (CC 100 and CC 110) demonstrated dose-dependent antibacterial activity. CuO NPs have a high surface-to-volume ratio that allows them to interact with the bacterial pathogen’s cell membrane across its surface, ultimately leading to the pathogen’s death [62]. To increase the surface responsiveness of NPs, electrical interactions created by CuO NPs with lower sizes and a bigger surface area were very useful. Furthermore, the increased surface area percentage immediately collaborates with the bacterium, resulting in improved bacterial engagement throughout the process. Cu and O, two essential components in NPs, significantly increase the antibacterial effectiveness of NPs with a large surface area [63,64]. Phytofabricated CuO NPs by mentha pulegium leaf/flower mixture reported by Alavi et al., 2021, and CuO NPs synthesized by electrochemical reduction method by Jadhav et al., 2011, demonstrated comparatively less or limited activity against Gram-positive and negative bacterial strains compared to CuO NPs’ activity in present work [65,66]. Additionally, Halbus et al., 2019 reported a novel type of modified CuO NPs which have been functionalized with GLYMO and 4-HPBA (CuO NPs/GLYMO/4-HPBA) to produce an antibacterial agent of much higher efficiency than bare CuO NPs [67]. Bio-synthesized CuO NPs showed versatile and higher antibacterial activity against various human and fish bacterial pathogens [68].

### 3.11. Safety Profiles of CuO NPs

*R. raphanistrum* seeds are grown on Murashige and Skoog agar containing CuO NPs at a concentration of 0–500 µg/mL in order to examine the efficient activity of CuO NPs on seed germination. CuO NPs did not cause any phenotypic changes in the seeds for the first four days; however, at 500 µg/mL, *R. raphanistrum*’s germination rate took longer to occur on day seven (Figure 11A). In comparison to untreated controls, seed germination and seedling growth are slightly decreased at the concentration of 100 µg/mL. Additionally, seedlings treated with CuO NPs at a concentration of 500 µg/mL had considerably shorter lengths than the untreated controls (Figure 11B). CuO NPs, particularly CC 100, displayed a comparable safety profile to that of biogenic copper nanoparticles and copper oxide-based nanocarriers as reported in earlier investigations [64,65]. Accordingly, it may be inferred from these side-by-side observations that the synthetic NPs are more secure for use in environmental applications.

## 4. Conclusions

In conclusion, CuO NPs are produced successfully utilizing an environmentally friendly method employing apple peel extract, and the NPs were evaluated against both Gram-positive bacterial pathogen *S. aureus* and Gram-negative bacterial pathogen *E. coli* to investigate its application in the medical sector. The bio-synthesized copper oxide nanoparticles inhibited the growth of *S. aureus* and *E. coli*. In order to strengthen our suggested hypothesis, we discovered that reasonable results of present research that prepared CuO NPs using microwave irradiation might be an effective alternative for treating *S. aureus-* and *E. coli*-associated infections. Additionally, seed germination toxic assay revealed that prepared CuO NPs support safe to use or disposal in the environment. Thus, CuO NPs might be promising ecofriendly and promising antibacterial agents based on their high dose safety and significant antibacterial efficacy against *S. aureus* and *E. coli*. Additionally, synthetic CuO NPs are more secure in the use of environmental applications.

## Figures and Tables

**Figure 1 micromachines-14-00456-f001:**
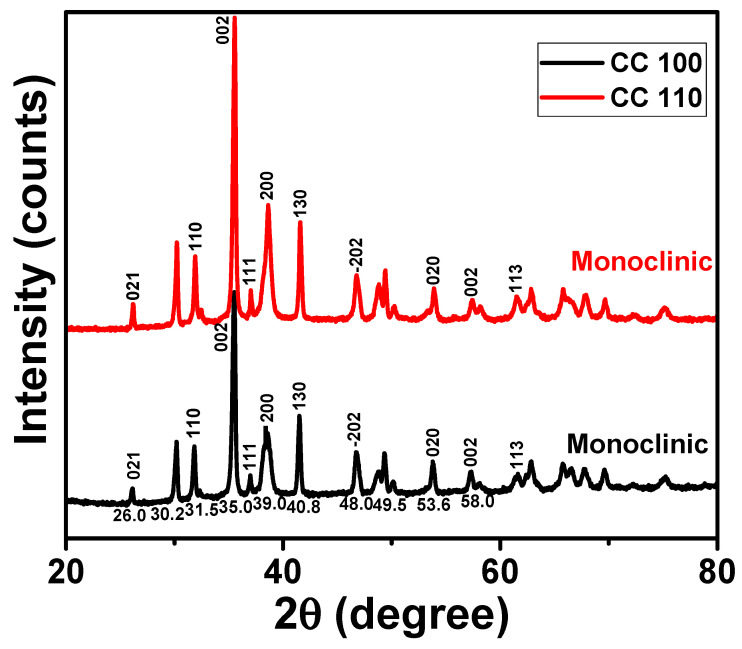
XRD patterns of CuO NPs.

**Figure 2 micromachines-14-00456-f002:**
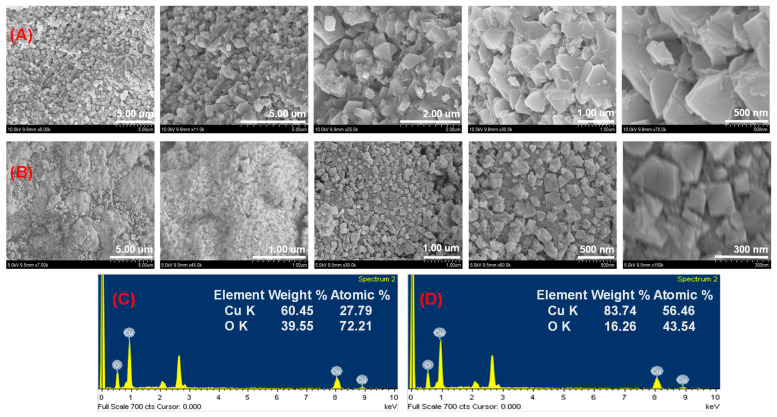
FE-SEM images of green synthetic CuO NPs (**A**) for the CC 100, (**B**) for the CC 110, EDX spectra of CuO NPs, (**C**) for the CC 100, and (**D**) for the CC 110.

**Figure 3 micromachines-14-00456-f003:**
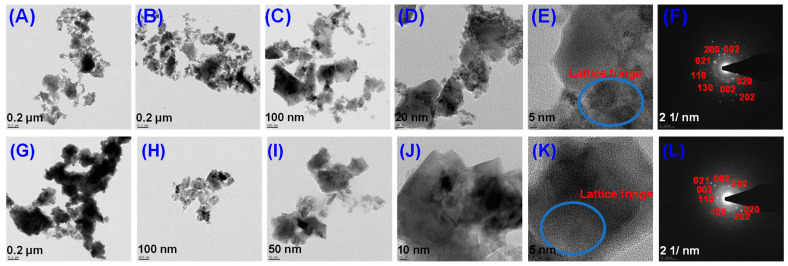
HR-TEM images of green synthetic CuO NPs (**A**–**E**) for CC 100, (**G**–**K**) for CC 110, and SAED patterns of CuO NPs (**F**) for CC 100, and (**L**) for CC 110.

**Figure 4 micromachines-14-00456-f004:**
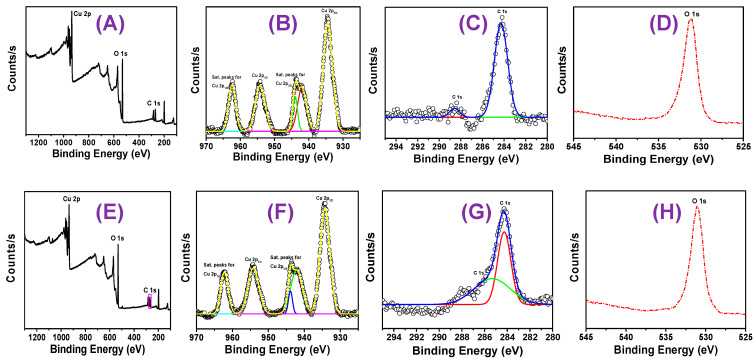
XPS of (**A**,**E**) survey scan spectrum of CuO NPs of CC 100, and CC 110, (**B**,**F**) Cu 2p of CC 100, and CC 110, (**C**,**G**) C 1s of CC 100, and CC 110, and (**D**,**H**) O 1s of CC 100, and CC 110.

**Figure 5 micromachines-14-00456-f005:**
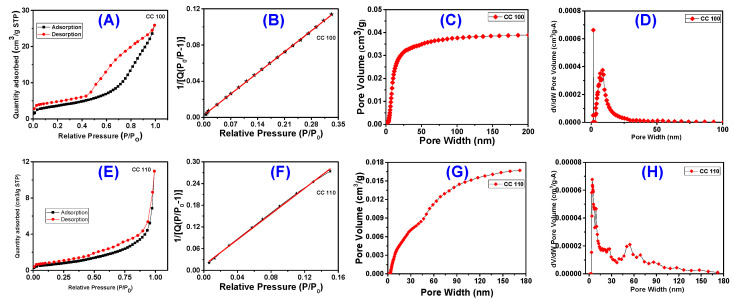
BET surface analysis with N2 gas adsorption-desorption isotherms of CuO NPs for CC 100 (**A**) and for CC 110 (**E**), surface area plot for CC 100 (**B**) and for CC 110 (**F**), BJH desorption pore size distribution for CC 100 (**C**) and for CC 110 (**G**), and differential pore volume plot for CC 100 (**D**) and for CC 110 (**H**).

**Figure 6 micromachines-14-00456-f006:**
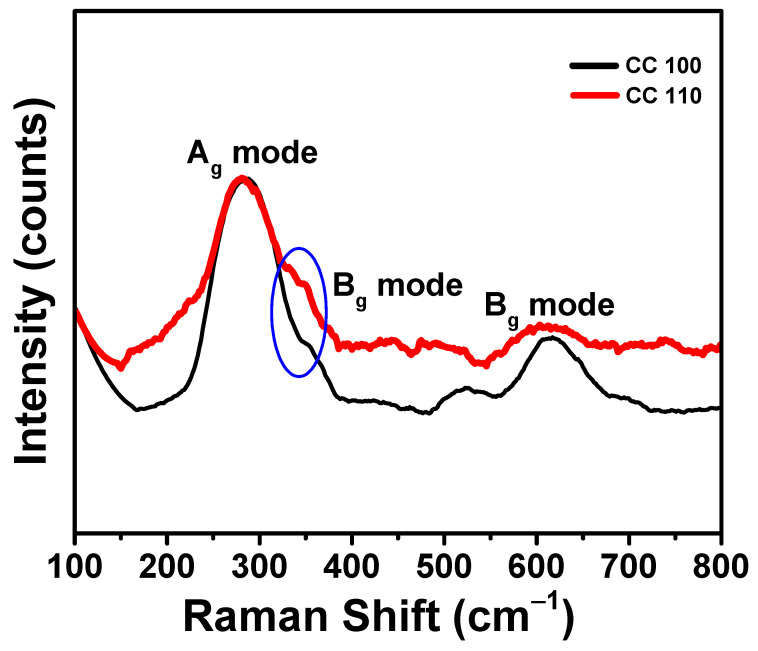
Raman spectra of CuO NPs.

**Figure 7 micromachines-14-00456-f007:**
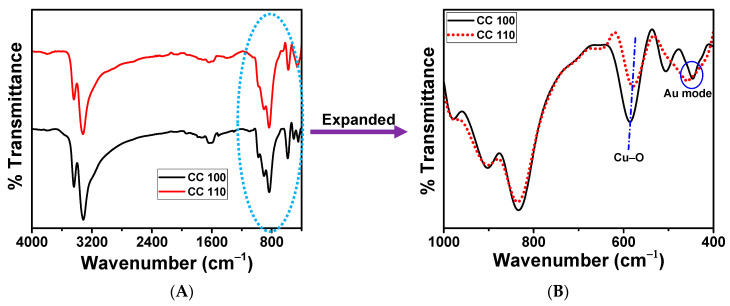
FT-IR spectra of CuO NPs for CC 100, and CC 110 (**A**), and expanded spectra (**B**).

**Figure 8 micromachines-14-00456-f008:**
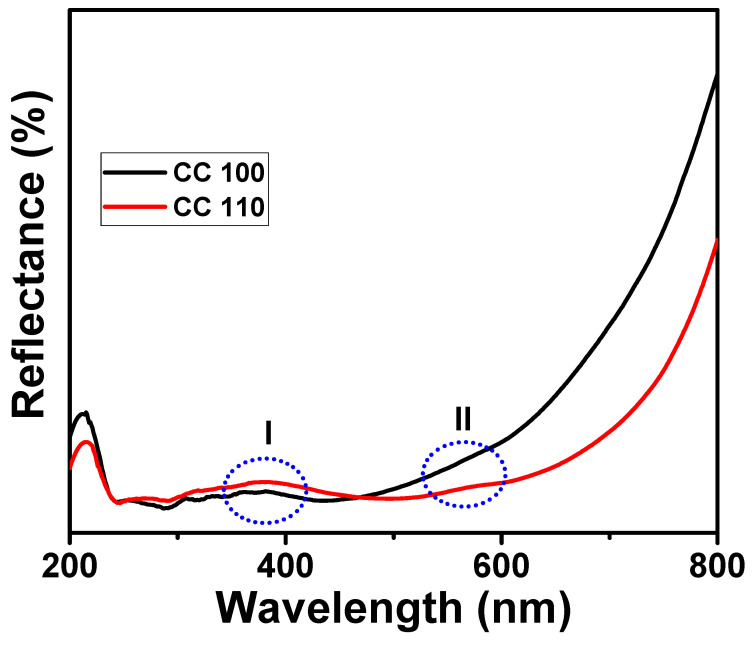
DRS spectra of CuO NPs.

**Figure 9 micromachines-14-00456-f009:**
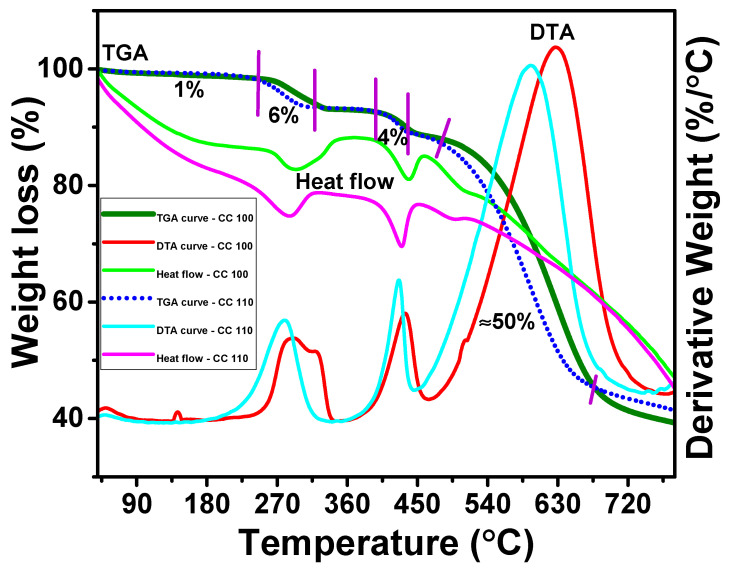
TGDTA of CuO NPs.

**Figure 10 micromachines-14-00456-f010:**
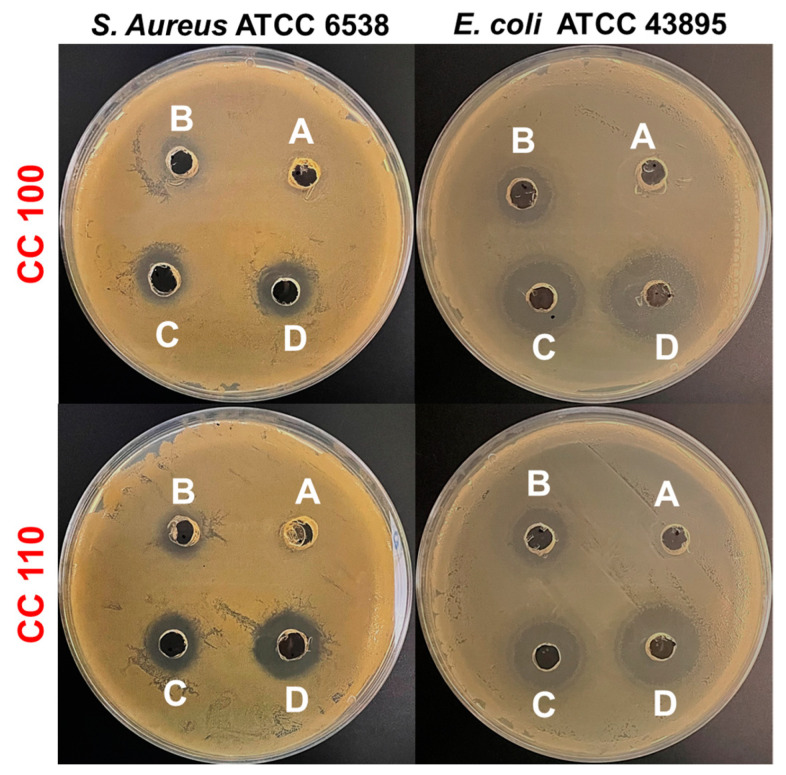
Antibacterial efficacy of CuO NPs with different concentrations (A, 0 µg/mL; B, 50 µg/mL; C, 100 µg/mL; D, 200 µg/mL).

**Figure 11 micromachines-14-00456-f011:**
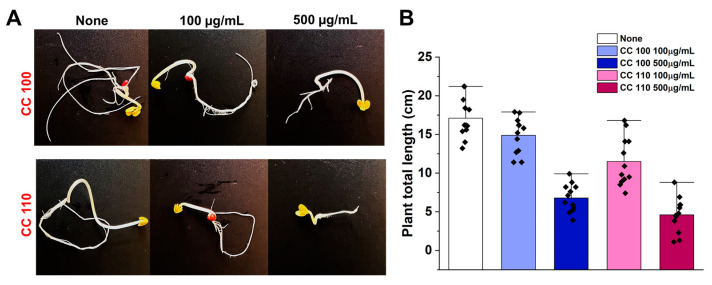
Effect of CuO NPs on seed germination of *R. raphanistrum* (**A**) bar graph demonstrates the effects of CuO NPs on seedlings’ length after seven days (**B**).

**Table 1 micromachines-14-00456-t001:** Binding energies of elements present in the NPs.

Name of the Sample	Cu 2p (eV)		O 1s (eV)	C 1s (eV)	
	Cu 2p_3/2_	Cu 2p_1/2_			
CC 100	934.88	954.48	531.18	284.18	288.38
CC 110	934.48	954.28	531.08	284.28	287.88

**Table 2 micromachines-14-00456-t002:** Surface area, pore volume, and pore size distribution of NPs by BET analysis.

Material	Surface Area (m^2^/g)		Pore Volume			Pore Size (nm)	
	t-Plot External Surface Area	Surface Area	t-Plot Micropore Volume (cm^3^/g)	BJH Adsorption Cumulative Volume of Pores (m^2^/g)	BJH Desorption Cumulative Volume of Pores (m^2^/g)	BJH Adsorption Average Pore Diameter (4V/A)	BJH Desorption Average Pore Diameter (4V/A)
CC 100	13.0291	12.9758	−0.000062	0.040396	0.040583	10.64	6.06
CC 110	3.6874	2.4368	−0.000483	0.017041	0.017084	17.08	13.03

**Table 3 micromachines-14-00456-t003:** Antibacterial efficacy of CuO NPs at different concentrations against *S. aureus* and *E. coli* by a zone of inhibition.

Bacterial Strains	Type of CuO NPs	Zone of Inhibition (mm)			
		Conc. of CuO NPs (µg/mL)			
		200	100	50	0
*S. aureus*	CC 100	16.0 ± 1.0	13.0 ± 0.9	9.0 ± 0.4	ND
	CC 110	16.1 ± 0.9	19.3 ± 0.4	11.1 ± 1.2	ND
*E. coli*	CC 100	23.1 ± 2.3	21.0 ± 0.8	15.0 ± 0.8	ND
	CC 110	23.2 ± 1.3	19.5 ± 0.5	14.4 ± 0.5	ND

## Data Availability

The datasets used or analyzed during the current study are available from the corresponding author upon reasonable request.

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
