# Peer review of "Novel Microwave Synthesis of Copper Oxide Nanoparticles and Appraisal of the Antibacterial Application"

_micromachines, 2023, doi:10.3390/mi14020456_

Round 1

Reviewer 1 Report

The authors bio-synthesized CuO NPs and characterized the NPs with several techniques supported each other. They showed the antibacterial property of the nanomaterial and investigated their effects on seed growing, which is good. The manuscript is publishable after minor revisions and  explanations. Please find my suggestions below.

1. In the introduction part (line 50-52), the authors explained the effects of CuNPs on plants. Explanation is a bit misleading. The authors claim that the CuNPs increase the ability of ROS production in plants but they should explain the mechanism of ROS production as a result of stress condition cause by CuNPs. Also, they should explain the activation of protection mechanism related with the ROS production.

2. The authors mention that the apple pees are very successful stabilizing agents in fabrication small sized NPs (page 3, line 124-125). There is no reference or any explanation about the role of apple pees to obtain especially small sized NPs. This part should be clarified by the authors.

3. In the result part, the authors mention about MIC values but we can not see any data interpretation about it. There is also no information about MIC test in Materials and Method part.

4. Why the authors did not perform the effect CuNPs on seed germination with using the same concentrations (25, 50, 200 ug/ml) they performed in agar diffusion test? An evaluation of comparison should be added in the manuscript.

Author Response

We are highly thankful to the Editor and the Reviewers for the insightful and constructive comments on our manuscript (Manuscript ID: micromachines-2178986) entitled “Novel microwave synthesis of copper oxide nanoparticles and appraisal of the antibacterial application”. As suggested by the Editor and the Reviewers, we have addressed the issues raised and modified the manuscript accordingly.

Dear Prof. Dr. Amber Zheng, Editor

We appreciate your kind consideration and constructive comments on our manuscript. As suggested by the editor and reviewers, we have markedly modified the manuscript. The editor’s and reviewer’s questions have been repeated in black text and our response follows in blue. In the revised manuscript, changes are in red.

Comments from the reviewers

Reviewer 1

The authors bio-synthesized CuO NPs and characterized the NPs with several techniques supported each other. They showed the antibacterial property of the nanomaterial and investigated their effects on seed growing, which is good. The manuscript is publishable after minor revisions and explanations. Please find my suggestions below.

We would like to thank the Reviewers for thorough reading of this manuscript and for the recommendations that have helped us improve the manuscript quality and scientific value.

  1. In the introduction part (line 50-52), the authors explained the effects of CuNPs on plants. Explanation is a bit misleading. The authors claim that the CuNPs increase the ability of ROS production in plants but they should explain the mechanism of ROS production as a result of stress condition cause by CuNPs. Also, they should explain the activation of protection mechanism related with the ROS production.

Author Responses : Dear Reviewer, the aim of this paper seems to be the production of NPs with the help of apple peel. For the introduction part, we just gave the introduction about the plants and their action to make nanoparticles. The proper references are also cited.

  1. The authors mention that the apple pees are very successful stabilizing agents in fabrication small sized NPs (page 3, line 124-125). There is no reference or any explanation about the role of apple pees to obtain especially small sized NPs. This part should be clarified by the authors.

Author Responses : The peel extract is the successful stabilizing agent and is evidenced by reference number 38. In the same way, we believed that the apple peel extract may be a successful stabilizer for making small-sized NPs and we finally achieved it. 

  1. In the result part, the authors mention about MIC values but we can not see any data interpretation about it. There is also no information about MIC test in Materials and Method part.

Author Responses : As suggested, we have interpreted our MIC results in the revised manuscript, and also the information about the MIC test was added in the material and method section.

  1. Why the authors did not perform the effect CuNPs on seed germination with using the same concentrations (25, 50, 200 ug/ml) they performed in agar diffusion test? An evaluation of comparison should be added in the manuscript.

Author Responses : Mainly the assay was carried out to determine the toxic effect of synthesized CuO NPs on seed germination of Raphanus raphanistrum. As highest MIC concentration of synthesized CuO NPs was 50 µg/mL. So, a toxicity assay was carried out at 2X MIC and 10X MIC concentrations of NPs. Therefore, seed germination assay was carried out at a higher concentration of NPs, and in comparison, to untreated controls, seedling growth are slightly decreased at the concentration of 100 µg/mL (2X MIC) but not showed any negative impact on R. raphanistrum seed germination.

Reviewer 2 Report

CuO nanoparticles could be synthesized from plant extracts and could be served as antibacterial agents. The topic of this manuscript is interesting. However, major revisions are required and the comments are given below.

1.     The authors need to provide more discussion on the innovation and significance of this work. Importantly, the authors should provide more discussion on the future perspectives of the presented research in the abstract, introduction, and conclusion to sublimate manuscript.

2.     Many inorganic nanoparticles could be synthesized from natural extracts and used as antioxidants, antibacterial materials, etc. More typical references are suggested to be cited for broad readers, e.g. Journal of Bioresources and Bioproducts 2021, 6 (1), 75-81; Journal of Bioresources and Bioproducts 2022, 7 (1), 1-13.

3.     Sentences like “Through the use of TEM microscopy and BET surface area” in line 16 and “Some of the NP-intensive industries are where it is possible for NPs to be released into the environment” in line 41 need to be revised. Please go through the whole manuscript to solve such issues.

4.     What is “ROS” in line 50? Please give the full name for abbreviations at the first time.

5.     The sentence “The synthesized NPs are characterized with the help of XRD, FE- SEM, HR-TEM, XPS, BET, and Raman spectral analysis” in line 65 is meaningless and could be deleted.

6.     “61.5 Å” for XRD pattern in line 102? The unit should be revised as “o”, that is “61.5 o”.

7.     The scale bars in the SEM images in Figure 2 and in TEM images in Figure 3 should be replaced by much clearer ones.

8.     How can t-Plot micropore volume be negative in Table 2? Please double check the data.

9.     Please pay attention to the writing of subscripts and superscripts

10.  How about the antibacterial efficacy of CuO NPs synthesized in this manuscript compared to that of counterparts reported in other papers?

Author Response

We are highly thankful to the Editor and the Reviewers for the insightful and constructive comments on our manuscript (Manuscript ID: micromachines-2178986) entitled “Novel microwave synthesis of copper oxide nanoparticles and appraisal of the antibacterial application”. As suggested by the Editor and the Reviewers, we have addressed the issues raised and modified the manuscript accordingly.

Dear Prof. Dr. Amber Zheng, Editor

We appreciate your kind consideration and constructive comments on our manuscript. As suggested by the editor and reviewers, we have markedly modified the manuscript. The editor’s and reviewer’s questions have been repeated in black text and our response follows in blue. In the revised manuscript, changes are in red.

Comments from the reviewers

Reviewer 2

CuO nanoparticles could be synthesized from plant extracts and could be served as antibacterial agents. The topic of this manuscript is interesting. However, major revisions are required and the comments are given below.

We would like to thank the Reviewers for their thorough reading of this manuscript and for the recommendations that have helped us improve the manuscript’s quality and scientific value.

  1. The authors need to provide more discussion on the innovation and significance of this work. Importantly, the authors should provide more discussion on the future perspectives of the presented research in the abstract, introduction, and conclusion to sublimate manuscript.

Author Responses : As per the reviewer’s suggestion, the future perspective of this work is provided in the abstract, introduction, and conclusion.

  1. Many inorganic nanoparticles could be synthesized from natural extracts and used as antioxidants, antibacterial materials, etc. More typical references are suggested to be cited for broad readers, e.g. Journal of Bioresources and Bioproducts 2021, 6 (1), 75-81; Journal of Bioresources and Bioproducts 2022, 7 (1), 1-13.

Author Responses : As per the suggestion, the specific references are now cited in the relevant places in the manuscript as the references are more suited.

  1. Sentences like “Through the use of TEM microscopy and BET surface area” in line 16 and “Some of the NP-intensive industries are where it is possible for NPs to be released into the environment” in line 41 need to be revised. Please go through the whole manuscript to solve such issues.

Author Responses : The sentence is revised and gone through the whole manuscript.

  1. What is “ROS” in line 50? Please give the full name for abbreviations at the first time.

Author Responses : ROS is abbreviated as Reactive oxygen species. Full name is provided now.

  1. The sentence “The synthesized NPs are characterized with the help of XRD, FE- SEM, HR-TEM, XPS, BET, and Raman spectral analysis” in line 65 is meaningless and could be deleted.

Author Responses : The sentence is rewritten now.

  1. “61.5 Å” for XRD pattern in line 102? The unit should be revised as “o”, that is “61.5 o”.

Author Responses : 61.5 Å” for XRD pattern with the miller indices of [113]. The unit is wrongly given in the manuscript, and now it is corrected. 

  1. The scale bars in the SEM images in Figure 2 and in TEM images in Figure 3 should be replaced by much clearer ones.

Author Responses : Scale bars are added in the SEM and TEM images.

  1. How can t-Plot micropore volume be negative in Table 2? Please double check the data.

Author Responses :  Data has been checked again. The unit is mistakenly provided, and now it is corrected (cm3/g).

  1. Please pay attention to the writing of subscripts and superscripts

Author Responses : As per the suggestion, the subscripts and superscripts are now checked and corrected.

  1. How about the antibacterial efficacy of CuO NPs synthesized in this manuscript compared to that of counterparts reported in other papers.

Author Responses : As suggested, we have compared the antibacterial efficacy of synthesized CuO NPs to previous reports on CuO NPs.

Round 2

Reviewer 2 Report

The manuscript has been revised according to the comments and could be accepted.